# Estimating Fractional Snow Cover in the Pan-Arctic Region Using Added Vegetation Extraction Algorithm

**Yuan Ma** [1,2,3], **Donghang Shao** [1,*], **Jian Wang** [1], **Haojie Li** [4], **Hongyu Zhao** [5] and **Wenzheng Ji** [1,3]

1   Key Laboratory of Remote Sensing of Gansu Province, Heihe Remote Sensing Experimental Research Station, Northwest Institute of Eco-Environment and Resources, Chinese Academy of Sciences, Lanzhou 730000, China
2   Key Laboratory of Disaster Prevention and Mitigation in Qinghai Province, Xining 810000, China
3   University of Chinese Academy of Sciences, Beijing 100049, China
4   College of Geography and Environmental Sciences, Northwest Normal University, Lanzhou 730070, China
5   State Key Laboratory of Earth Surface Processes and Resource Ecology, Beijing Normal University, Beijing 100875, China
*   Correspondence: shaodonghang@lzb.ac.cn

**Abstract:** Snow cover is an essential indicator of global climate change. The composition of the underlying surface in the Pan-Arctic region is complex; forest and other areas with high vegetation coverage have a significant influence on the retrieval accuracy of fractional snow cover (FSC). Therefore, to explore the impact of vegetation on the extraction of the FSC algorithm, this study developed the normalized difference vegetation index (NDVI)-based Bivariate Linear Regression Model (BV-BLRM) to calculate the FSC. Then, the overall accuracy of the model and its changes under different classification conditions were evaluated and the relationship between the accuracy improvement and different underlying surfaces and elevations was analyzed. The results show that the BV-BLRM model is more accurate than MODIS's traditional univariate linear algorithm for FSC (MOD-FSC) in each underlying surface. Overall, regarding the accuracy of the BV-BLRM model, the RMSE is 0.2, MAE is 0.15, and accuracy is 28.6% higher than the MOD-FSC model. The newly developed BV-BLRM model has the most significant improvement in the accuracy of FSC retrieval when the underlying surface has high vegetation coverage. Under different classification accuracies, the accuracy of BV-BLRM model was higher than that of MOD-FSC model, with an average of 30.5%. The improvement of FSC extraction accuracy by the model is smaller when the underlying surface is perpetual snow zone, with an average of 12.2%. This study is applicable to the scale mapping of FSC in large areas and is helpful to improve the FSC accuracy in areas with high vegetation coverage.

**Keywords:** fractional snow cover; BV-BLRM model; vegetation; classification of validation

## 1. Introduction

Snow cover is essential to the cryosphere and global climate system. About 98% of the seasonal snow cover on earth is in the northern hemisphere [1,2]. In winter, the maximum snow cover is about $47 \times 10^6$ km$^2$, accounting for 60% of the land area in the northern hemisphere [3–6]. With global warming, climate change is becoming increasingly evident, the frequency of extreme climate events is increasing, and the snow cover area is also changing [7]. Fractional snow cover (FSC) at a sub-pixel scale as a substitute for the snow area can provide a reference for a more accurate estimation of the snow cover area.

The FSC retrieved from optical remote sensing data can be divided into empirical model/semi-empirical model, mixed pixel decomposition, and machine learning models. An empirical/semi-empirical model is used to establish the relationship between the surface reflectance and FSC by the spectral mixing algorithm and statistical method so as to retrieve the FSC [8–11]. The mixed pixel decomposition calculates the FSC by solving the proportion or abundance of the snow end elements in the pixel. The mixed pixel decomposition method is mainly based on spectral analysis [12–17]. The machine

learning algorithm prepares FSC by training the relationship between FSC and various factors (such as surface reflectance, vegetation, etc.). Specifically, the machine learning algorithm of artificial neural networks (ANNs) are used for snow recognition and snow mapping [18–20], and then the FSC is prepared by combining ANNs with wavelet analysis, support vector machines (SVMs), and multivariate adaptive region splines (MARS) [21–23]. The hybrid pixel decomposition model and machine learning model algorithm have poor computational complexity and universality; there are still significant uncertainties in the mapping of FSC on the complex underlying surfaces [17,24]. Due to their relatively low computational complexity, empirical/semi-empirical models have the advantages of fast running speed, easy data acquisition, and accuracy in meeting basic needs in large-scale and long-term time series FSC mapping [25,26]. Most empirical/semi-empirical models calculate FSC by establishing the regression relationship between the normalized difference snow index (NDSI) and other variables. Representative studies include a piecewise linear regression model based on a semi-empirical linear FSC calculated by NDSI considering the sparsity of snow cover [18,23]. The FSC's semi-empirical exponential regression model integrates geographical location, temperature, terrain, reflectance, vegetation, and snow cover index [27].

The reflectance of snow in the visible band of forest area is vulnerable to a decrease due to the influence of vegetation cover, and the NDSI calculated in this area is usually lower than that in other areas. So, the empirical/semi-empirical model obtained from the regression of NDSI and other variables is used to retrieve the FSC, and the snow cover area in forest areas is often seriously underestimated [28–30]. Comprehensive snow mapping in forest areas considering NDSI and NDVI has been widely used [9,28]. Representative studies on snow cover mapping in forest areas include the linear spectral hybrid model SnowFrac, which estimates the proportion of snow cover area through spectral decomposition, but it requires a high-precision forest cover map as prior knowledge [8,31]. The model is cannot easily retrieve the proportion of snow cover in an extensive range. The SCAmod FSC retrieval model based on single-channel reflectance [10,32] has achieved high accuracy when used to map snow cover areas in the Finnish plain. However, in mountainous regions with prominent surface heterogeneity, single-channel reflectance retrieval is bound to have the defects of a single observation angle and less time.

To quantify the effect of vegetation on the extraction accuracy of FSC, in this paper, we proposed a retrieval model of FSC based on vegetation characteristics and NDSI. This algorithm aims at viewable snow. Considering the distribution characteristics of snow cover in vegetation and non-vegetation areas, compared with the MODIS algorithm, this study adds vegetation information and constructs a proportion regression model of FSC in combination with NDVI and NDSI segments. Landsat 8 Surface Reflectance (Landsat 8 SR) data are used as the "reference true data" to verify the accuracy of the regression model. In addition, the retrieval results are compared with the conventional linear univariate algorithm for FSC given in MOD10A1 V6 Snow Cover Daily (MOD10A1 SCD). The accuracy of FSC estimation by the BV-BLRM model under different underlying surfaces and elevations was compared, and the influence of vegetation on FSC was analyzed.

## 2. Study Area and Data

### 2.1. Study Area

The study area is located in the land area north of a 45° north latitude (from now on, referred to as the Pan-Arctic region), including Asia, Europe, and North America, with a cold climate and extensive snow coverage (Figure 1). The land area covers countries such as Russia, the United States, Canada, Denmark, Norway, Iceland, Sweden, and Finland. In this paper, 38 images were selected as training samples, and 19 images were selected as verification samples. The spatial distribution of training samples is shown in Figure 1. The red box represents the training sample areas; the black box represent the verification sample areas.

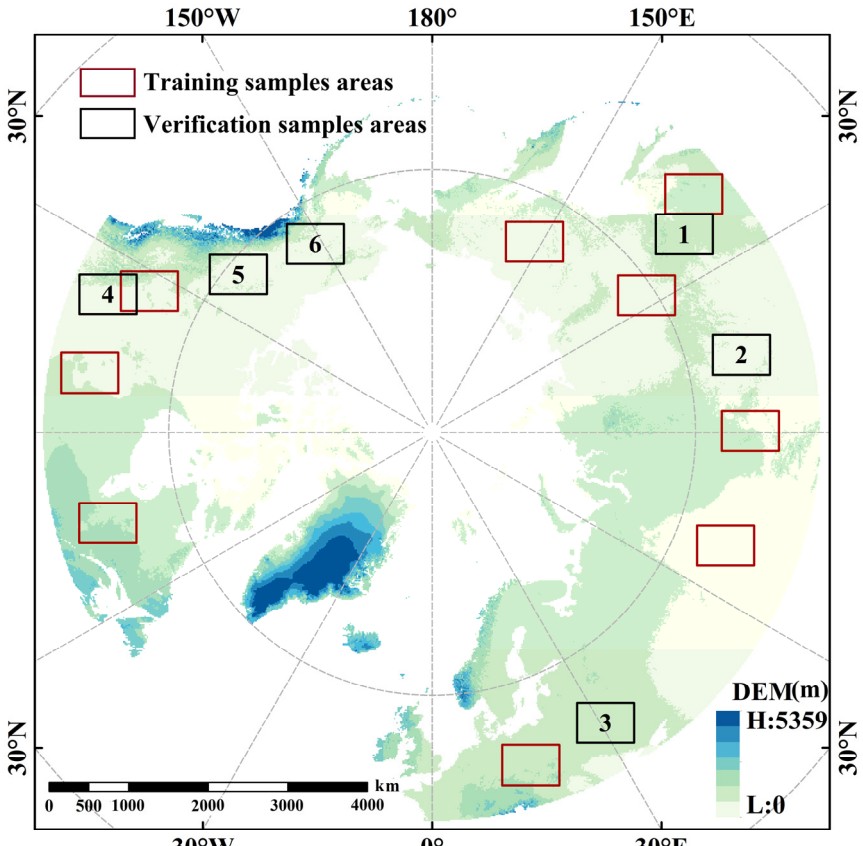

**Figure 1.** Elevation profile of the Pan-Arctic region (above 45° latitude) and spatial distribution of sample areas (the red box is the spatial distribution of sample areas used for model training and the black box is the spatial distribution of sample areas used for model verification).

### 2.2. Data and Pretreatment

The main data used in this study include DEM data, Landsat 8 SR, MOD09GA V6, MOD10A1 V6 (MODIS snow product), and MCD12Q1 global annual land cover type data. Among them, Landsat 8 SR, MOD09GA, and MOD10A1 data have been atmospherically corrected. The research was carried out on the GEE platform.

#### 2.2.1. DEM

DEM (SRTM90) data were collected from USGS (United States Geological Survey, USGS, http://www.usgs.gov/, accessed on 10 September 2019). The spatial resolution is 90 m, and the geographic coordinate system is WGS 84. DEM data are mainly used for altitude gradient classification.

#### 2.2.2. Landsat 8 SR (TFSC)

The data come from the USGS and the effect of the atmosphere on surface reflectance has been removed. It has a spatial resolution of 30 m, a temporal resolution of 16 days, and a projection of WGS84 UTM. Since the sun azimuth angle and the terrain's elevation have significant effects on the surface reflectance, this paper used the terrain correction method to remove the surface reflectance effect. In this paper, the VECA [33] method was used to conduct terrain correction for Landsat 8 SR data (Equations (1) and (2)):

$$\cos i = \cos\left(\theta_{slop}\right)\cos(\theta_{sz}) + \sin\left(\theta_{slop}\right)\sin(\theta_{sz})\cos\left(\theta_{sa} - \theta_{aspect}\right) \tag{1}$$

$$\frac{R_{corr}}{R_{uncorr}} = \frac{L_a}{m \times \cos i + b} \tag{2}$$

where $\theta_{slop}$ is the slope, $\theta_{aspect}$ is the exposure, $\theta_{sz}$ is the zenith angle, $\theta_{sa}$ is the azimuth, $L_a$ is the average reflectance of an uncorrected image, $m$ and $b$ are the correction coefficient, $R_{corr}$ is corrected data, and $R_{uncorr}$ is uncorrected data.

After terrain correction, the Landsat 8 SR data were used to calculate NDVI (Equation (3)) and prepare the TFSC dataset.

$$NDVI = \frac{NIR_{0.86} - Red_{0.65}}{NIR_{0.86} + Red_{0.65}} \tag{3}$$

where $Red_{0.65}$ is the reflectance of red band of Landsat 8 SR, and $NIR_{0.86}$ is the reflectance of near-infrared band of Landsat 8 SR.

When preparing TFSC, the SNOMAP algorithm was used to prepare the binary image of the snow cover. It was then aggregated into a 500 m resolution FSC dataset, which was used as the input data of the multi-parameter weight regression model. The SNOMAP algorithm identifies pixels with NDSI $\geq$ 0.4, B5 > 0.11, and B3 > 0.1 as snow pixels in non-forest areas and uses the dynamic NDSI-NDVI method to identify snow in forest areas. In the forest area, pixels with NDSI $\geq$ 0.2 and NDVI > 0.1 are also recognized as snow pixels. The calculation formula of NDSI is as Equation (4) [28]:

$$NDSI = \frac{Green_{0.56} - SWIR_{1.61}}{Green_{0.56} + SWIR_{1.61}} \tag{4}$$

where $Green_{0.56}$ is the green light reflectance of Landsat 8 SR, and $SWIR_{1.61}$ is the shortwave infrared reflectance of Landsat 8 SR.

In the aggregation method, the 30 m-resolution binary snow data were scaled up to 500 m-resolution FSC data, which could be considered the TFSC. The formula of aggregation is as outlined in Equation (5).

$$FSC_i = \frac{s}{n} = \frac{\sum_{k=1}^{s} 1}{[500/30]^2} \tag{5}$$

where $FSC_i$ is the FSC value of $i$ pixel after aggregation to 500 m resolution, $s$ is the sum of FSC values in the pixel, and $n$ is the number of 30 m pixels included in 500 m pixels.

### 2.2.3. MODIS Data

MOD09GA MODIS/Terra satellites for remote sensing data generated by the world day by day the surface reflectivity data (https://search.earthdata.nasa.gov/search, accessed on 10 September 2019). The spatial resolution of the dataset is 500 m and 1000 m, with sinusoidal projections. This dataset has been atmospherically corrected, and its projection is converted to WGS 84 UTM for subsequent calculations in this paper. NDVI and NDSI were calculated by MOD09GA and used as input data in the regression model together with the surface reflectance of MOD09GA.

Equation (6) is used to calculate MFSC. The estimated results can be regarded as FSC calculated by the MODIS empirical model, which can be compared with other models [25].

$$FSC = 1.45 \times NDSI - 0.01 \tag{6}$$

### 2.2.4. Categorical Data

Terra and Aqua obtain MCD12Q1 (1 January 2008) and binary supervised classification combined global surface coverage type data year by year (https://search.earthdata.nasa.gov/, accessed on 10 September 2019), which has been proven to have high classification accuracy. The data will be used as auxiliary data to evaluate the accuracy of the BV-BLRM algorithm under different underlying surfaces. Its spatial resolution is 500 m with sinusoidal projection. Considering the complexity of the underlying character of the study area, the land cover type data were used as the primary data for zoning verification. To reduce the complexity and improve the efficiency, this study classified the differences

between land cover types and DEM according to the vegetation height and coverage, as shown in Table 1 below. Among them, the proportion of areas covered by vegetation was as high as 61.2%. In the study area, 84.4% of the elevation is between 0 and 1000 m, and the proportion of the area higher than 5000 m is close to 0%. This classification will be removed in the subsequent study. It analyzes the relationship between vegetation, elevation, and FSC accuracy. A total of 15 training sample images are in vegetation areas and 23 training sample images are in non-vegetation areas (Figures 1 and 2). Table 2 shows the validation of metadata information for the dataset.

**Table 1.** Reclassification criteria.

| Category Name | Data Source | Classification Standard | Proportion (%) |
| --- | --- | --- | --- |
| Forest | | LC_Type1 (1,2,3,4,5) | 2.2% |
| Bushwood | | LC_Type1 (6,7) | 17.2% |
| Grassland | | LC_Type1 (8,9,10) | 16.9% |
| Cropland | MCD12Q1 | LC_Type1 (12,14) | 24.9% |
| Nudation | | LC_Type1 (11,13,16) | 17.8% |
| Snow | | LC_Type1 (15) | 13.8% |
| Water | | LC_Type1 (17) | 7.2% |
| 1000 m | | $0 \text{ m} \leq \text{elevation} < 1000 \text{ m}$ | 85.5% |
| 2000 m | | $1000 \text{ m} \leq \text{elevation} < 2000 \text{ m}$ | 12.1% |
| 3000 m | | $2000 \text{ m} \leq \text{elevation} < 3000 \text{ m}$ | 2.2% |
| 4000 m | SRTM90_V4 | $3000 \text{ m} \leq \text{elevation} < 4000 \text{ m}$ | 0.1% |
| 5000 m | | $4000 \text{ m} \leq \text{elevation} < 5000 \text{ m}$ | <0.1% |
| >5000 m | | $\text{elevation} \geq 5000 \text{ m}$ | 0% |

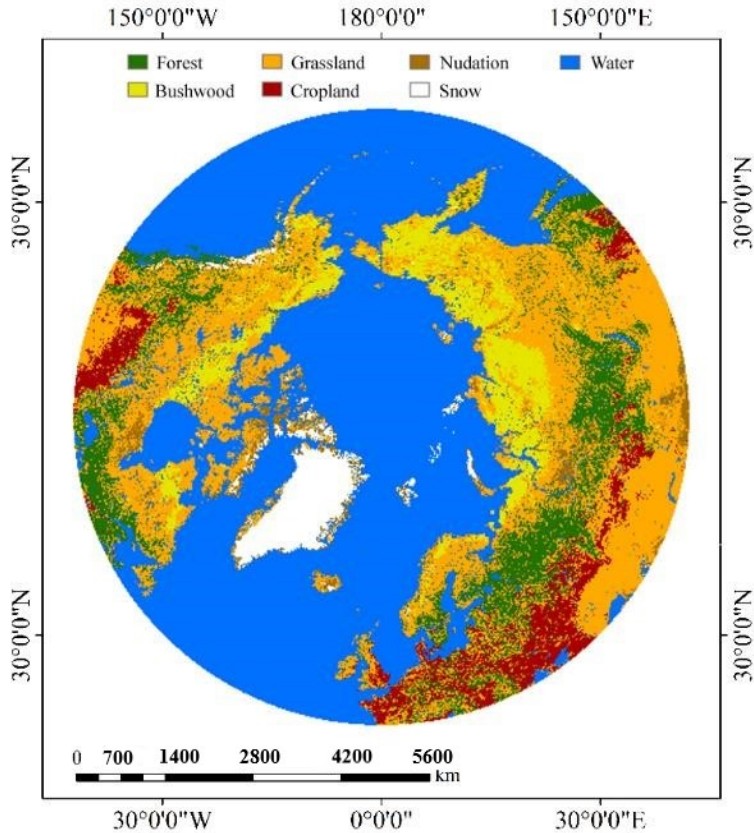

**Figure 2.** Result map of MCD12Q1 land cover type reclassification.

**Table 2.** Validation of metadata information for the dataset.

| Date | Cloud Cover (%) | Partition Labeled | Type of Underlying Surface |
|---|---|---|---|
| 20181006 | 0.45 | 1 | Grassland |
| 20151010 | 0.09 | 1 | Forest |
| 20181018 | 0.88 | 1 | Cropland |
| 20140325 | 1.04 | 2 | Grassland |
| 20151216 | 0.2 | 2 | Forest |
| 20140203 | 0.02 | 2 | Nudation |
| 20171115 | 0.45 | 3 | Forest |
| 20161229 | 0.98 | 3 | Cropland |
| 20150226 | 0.52 | 3 | Nudation |
| 20161129 | 0.02 | 4 | Cropland |
| 20170103 | 0.15 | 4 | Cropland |
| 20151231 | 0.19 | 4 | Grassland |
| 20180224 | 0.52 | 4 | Grassland |
| 20160228 | 0.86 | 5 | Forest |
| 20151101 | 0.22 | 5 | Grassland |
| 20181003 | 0.98 | 5 | Nudation |
| 20170928 | 1.33 | 6 | Snow |
| 20151001 | 1.62 | 6 | Bushwood |
| 20180320 | 1.95 | 6 | Nudation |

## 3. Methods

In this paper, we developed the BV-BLRM piecewise linear regression model with NDVI, which is suitable for preparing large-scale, long time series and high-resolution FSC data. We developed a large-scale FSC calculation model BV-BLRM algorithm. At the same time, the FSC data obtained by the SNOMAP algorithm from Landsat 8 SR data were used as the actual-value FSC (TFSC) [34], MODIS/Terra Surface Reflectance Daily L2G Global 1 km and 500 m SIN Grid V006 (MOD09GA) data through the BV-BLRM model were used to calculate FSC (VFSC), and the FSC calculated by the MOD-FSC model was used as the reference FSC (MFSC). Then, the MODIS Terra/Aqua Land Cover Type Yearly L3 Global 500 m SIN Grid V006 (MCD12Q1) data were reclassified into seven different underlying surface types according to the difference in vegetation height and coverage. DEM was divided into seven classes according to different sizes. Finally, the classification verification of the calculated FSC was carried out to illustrate further the error distribution law of FSC in the Pan-Arctic region.

(1)   The terrain-corrected Landsat 8 SR data are calculated by using the SNOMAP algorithm to obtain the FSC data with a resolution of 30 m, then the 30 m resolution FSC data were aggregated into 500 m resolution, and the NDVI and NDSI values were calculated from MOD09GA data (Figure 3).

(2)   The FSC, NDSI, and NDVI data in step 1 were used to determine the coefficients of the BV-BLRM model by least square fitting to obtain the BV-BLRM model.

(3)   In the verification, in step 1, the 500 m resolution FSC obtained by Landsat 8 SR is used as the true value FSC (TFSC), the MOD09GA data is calculated by the BV-BLRM model to obtain VFSC, and the MOD-FSC model is calculated to obtain MFSC, where VFSC is the verification data and MFSC is the comparison data.

(4)   The underlying surface and altitude are divided into 12 categories, and the errors between VFSC and TFSC, MFSC, and TFSC are analyzed, respectively.

(5)   Based on the distribution law of error between different underlying surfaces and different altitudes, combined with the distribution law of vegetation at different altitudes, the influence of vegetation on the extraction of the snow area proportion is further analyzed.

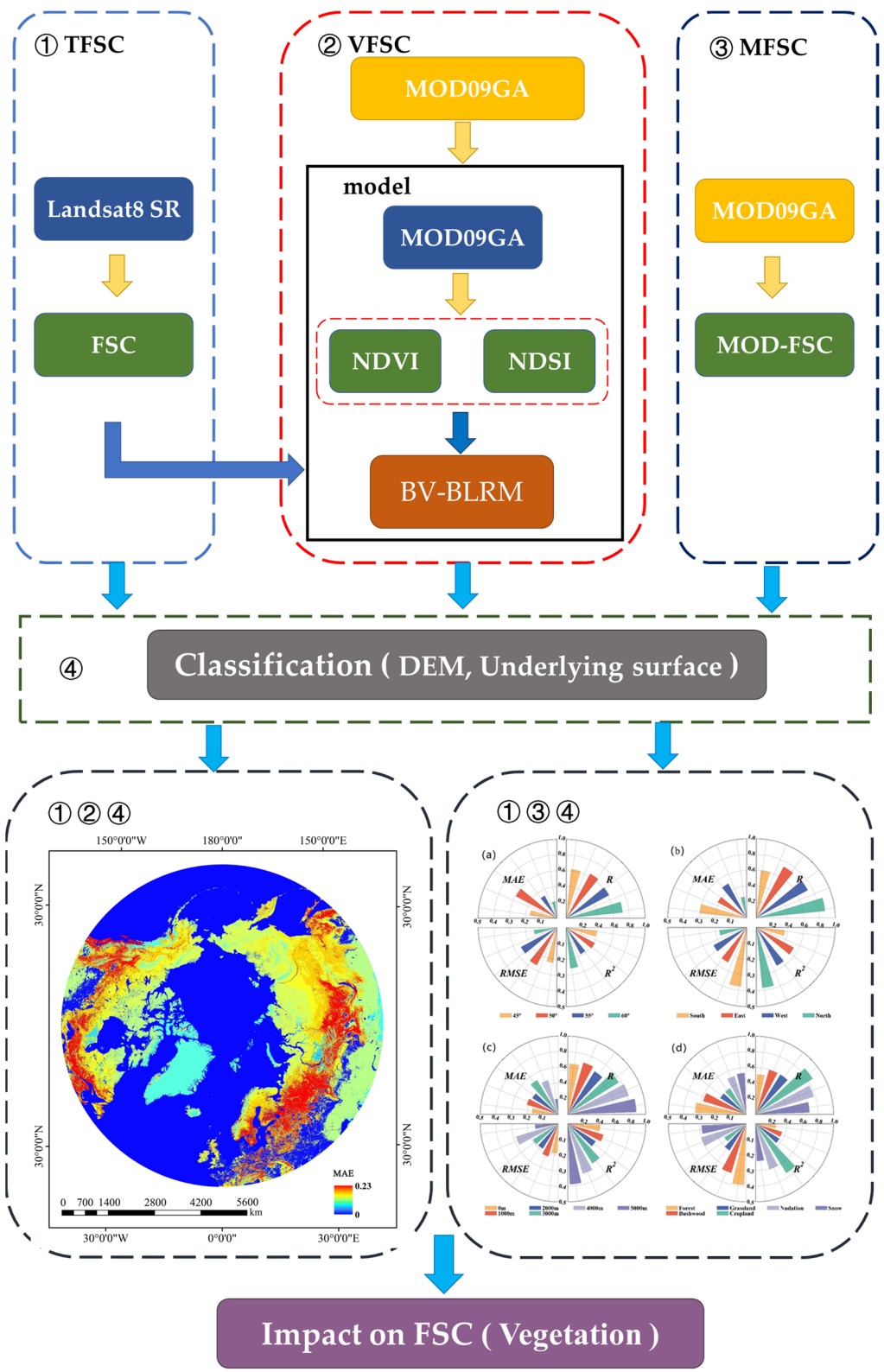

**Figure 3.** Overview flow chart (① represents true FSC (TFSC); ② the black box shows the establishment of the BV-BLRM model and represents new FSC (VFSC); ③ represents reference FSC (MFSC); ④ according to the different categories, they are DEM and the underlying surface).

### 3.1. Building the BV-BLRM Model

By constructing a linear equation between NDVI and NDSI and FSC, the BV-BLRM model is developed.

$$FSC = \begin{cases} a_1 \cdot NDSI + a_2 \cdot NDVI + a_3, NDVI > \text{m} \\ b_1 \cdot NDSI + b_2, NDVI \leq \text{m} \end{cases} \tag{7}$$

where $a_1$, $a_2$, $a_3$, $b_1$, and $b_2$ are the regression coefficients, and m is the NDVI value that distinguishes the vegetation area and non-vegetation area. Segmented modeling was carried out according to NDVI. In this paper, the linear least square method is used to solve the linear regression model, so Equation (7) can be expressed as:

$$\min_x \frac{1}{2} \|FSC - AX\|^2 \tag{8}$$

where $A = (a_1, a_2, a_3)|(b_1, b_2)$, $X = (NDSI, NDVI, 1)^T|(NDSI, 1)^T$.

Specifically, Landsat 8 SR data with snow and a cloud cover of less than 2% are selected and preprocessed as FSC data with a 500 m resolution. The preprocessing method is shown in Section 2.2. Then, the corresponding MOD09GA data were selected, and NDVI and NDSI were calculated. Finally, Equation (8) was used for linear fitting.

### 3.2. Precision Evaluation

The FSC of long-time series were classified and verified by randomly selected images based on Google Earth Engine (GEE).

In this paper, the correlation coefficient (R), coefficient of determination ($R^2$), root mean square error (RMSE), and mean absolute error (MAE) were used as model accuracy evaluation factors [35].

$$R(x, y) = \frac{\sum_{i=1}^{n} (x_i - \overline{x_i})(y_i - \overline{y_i})}{\left(\sum_{i=1}^{n} (x_i - \overline{x_i})^2\right)^{\frac{1}{2}} \left(\sum_{i=1}^{n} (y_i - \overline{y_i})^2\right)^{\frac{1}{2}}} \tag{9}$$

$$R^2(x, y) = \frac{\sum_{i=1}^{n} (x_i - \overline{y_i})^2}{\sum_{i=1}^{n} (y_i - \overline{y_i})^2} \tag{10}$$

$$RMSE(x, y) = \left(\frac{\sum_{i=1}^{n} (x_i - y_i)^2}{n}\right)^{\frac{1}{2}} \tag{11}$$

$$MAE(x, y) = \frac{\sum_{i=1}^{n} |x_i - y_i|}{n} \tag{12}$$

where $x_i$ is the observed data (VFSC or MFSC), $y_i$ is the truth value (TFSC), $\overline{x_i}$ is the average of the observations, $\overline{y_i}$ is the average of the truth values, and $n$ is the indicates the number of samples.

For Landsat 8 SR data, the corresponding MOD09GA surface reflectance data were selected through time registration, and the spatial resolution, projection, and spatial range were unified. VFSC and MFSC were obtained. The binary snow data provided by MOD10A1 were used to mask the VFSC and MFSC data; the data of VFSC and MFSC outside the scope of the snow cover are invalid values. The invalid values are assigned as null values and will not participate in subsequent calculations. The masked VFSC, MFSC, and TFSC data are generated into a new dataset containing three bands, where bands 1–3 are VFSC, MFSC, and TFSC, respectively. MCD12Q1 reclassification and DEM classification were carried out, the new dataset was classified, and the accuracy was evaluated. VFSC and MFSC were compared with TFSC, and four error evaluation functions (R, $R^2$, MAE, RMSE) were used to assess the accuracy.

## 4. Results

### *4.1. BV-BLRM Model Results and Validation*

#### 4.1.1. BV-BLRM Model Results

The multiple linear regression model was established according to the selected training samples, and the linear regression model was established for 3.4 million effective snow pixels after removing the cloud pixels. The method to remove cloud pixels is as follows: select the part with cloud amount less than 2% when selecting the Landsat 8 SR image; the cloud amount data are from Landsat 8 SR. According to the statistical land cover data, the surface land cover is classified into two categories: vegetation and non-vegetation areas. Then, extract the NDVI corresponding to these two regions; the NDVI data are calculated by MOD09GA. According to statistics of NDVI of vegetation and non-vegetation areas, 0.2 is the value that can best distinguish these two areas. It is considered that when NDVI is greater than 0.2, there is vegetation cover; less than or equal to 0.2 is non-vegetation. When FSC solutions have an NDVI greater than 0.2, $a_1 = 1.05$, $a_2 = -0.08$, $a_3 = 0.1$. When NDVI is less than or equal to 0.2 and greater than 0, $b_1 = 1.06$, $b_2 = 0.19$. Then, Equation (7) can be expressed as Equation (13):

$$FSC = \begin{cases} 1.05 \cdot NDSI - 0.08 \cdot NDVI + 0.1, NDVI > 0.2 \\ 1.06 \cdot NDSI + 0.19, NDVI \leq 0.2 \end{cases} \tag{13}$$

#### 4.1.2. BV-BLRM Model Validation

In this study, 19 groups of images (Table 2, eligible Landsat 8 SR data and corresponding MOD09GA data) were selected to verify the accuracy of FSC preparation by the BV-BLRM algorithm. R, $R^2$, RMSE, and MAE were used as evaluation indexes, and the average value of all test samples was calculated. The partition labeling of the spatial distribution is shown in Figure 1.

When comparing the MOD-FSC model and BV-VLRM model to calculate FSC, it is found that VFSC is closer to TFSC in the low value area than MFSC (Figure 4a). In the high-value area, VFSC is closer to TFSC than MFSC (Figure 4b,d). VFSC shows that BV-BLRM model significantly improves the estimation of FSC in low-value areas (Figure 4c). VFSC is closer than the MFSC low-value area, and MFSC overestimates FSC significantly (Figure 4f). In the case of underlying surfaces of different types of vegetation, the RMSE and MAE of VFSC are improved to varying degrees (Table 3).

**Table 3.** Precision verification results of FSC prepared by different models (different vegetation categories).

| Surface Coverage Category | VFSC | | MFSC | |
|---|---|---|---|---|
| | **RMSE** | **MAE** | **RMSE** | **MAE** |
| Mixed forest | 0.298 | 0.296 | 0.483 | 0.515 |
| Deciduous broad-leaved forest | 0.332 | 0.488 | 0.502 | 0.54 |
| Medium sparse forest | 0.325 | 0.446 | 0.628 | 0.614 |
| Sparse forest | 0.255 | 0.313 | 0.511 | 0.443 |
| Nudation | 0.277 | 0.202 | 0.323 | 0.298 |
| Cultivated land | 0.219 | 0.196 | 0.286 | 0.236 |
| Grassland | 0.297 | 0.311 | 0.349 | 0.369 |

Table 4 shows the average accuracy of the BV-BLRM model and MOD-FSC model. The results show that, compared with the MOD-FSC model, R and $R^2$ of the BV-BLRM model are improved, and RMSE and MAE are decreased. This indicates that the introduction of NDVI significantly impacts snow area proportion algorithm extraction.

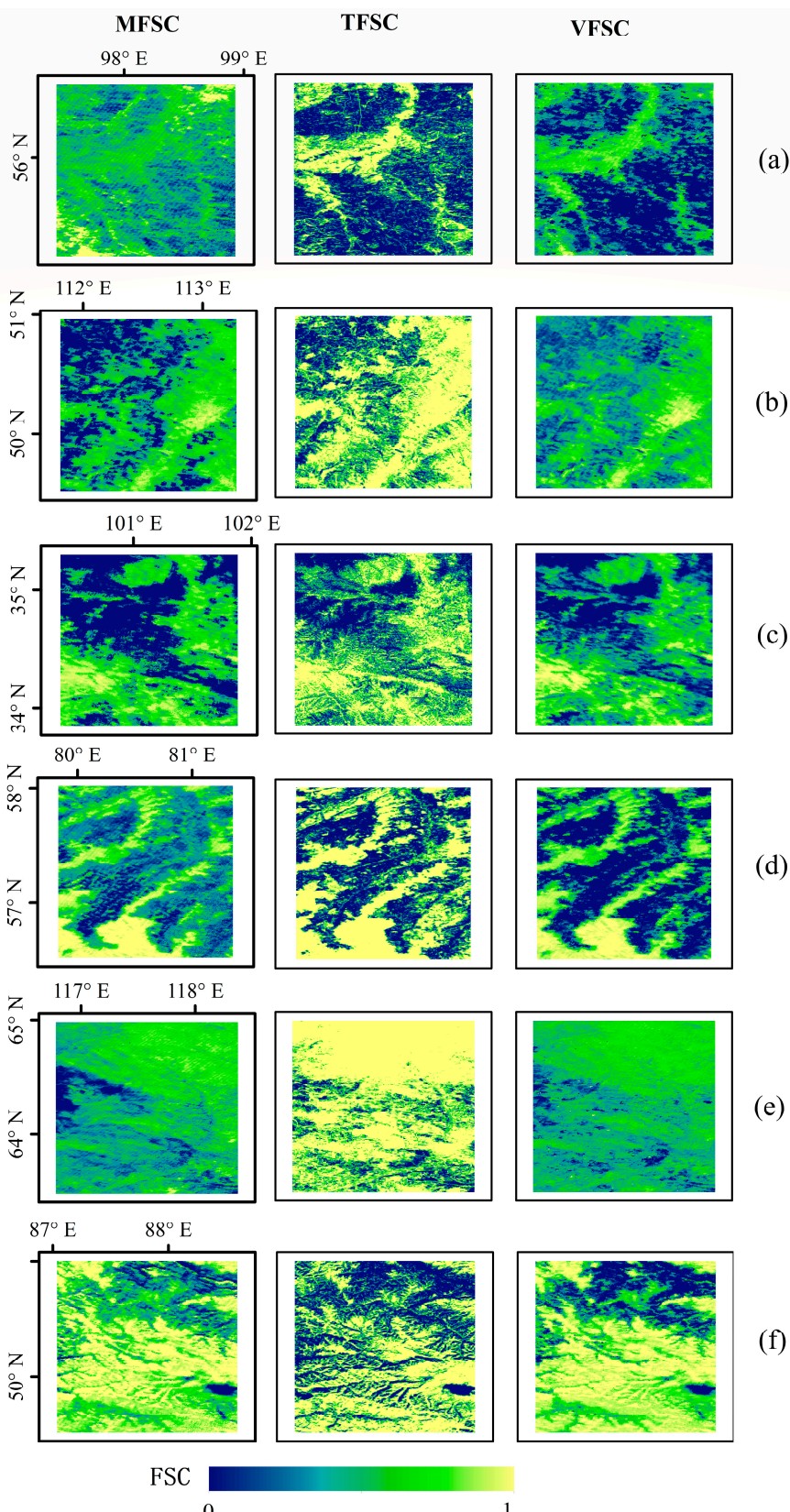

**Figure 4.** Results of FSC preparation by different models (MFSC is the FSC product of MOD-FSC, TFSC is the "true value" FSC of Landsat 8, and VFSC is the FSC of BV-BLRM model. (**a–f**) stands for different numbers).

**Table 4.** Accuracy of FSC preparation by different models.

| Model | R | $R^2$ | RMSE | MAE |
|---|---|---|---|---|
| BV-BLRM | 0.72 | 0.52 | 0.2 | 0.15 |
| MOD-FSC | 0.62 | 0.38 | 0.29 | 0.21 |

BV-BLRM is used to prepare FSC data in a large area. Figure 5 shows the application results of the model in the Pan-Arctic. For daily FSC data, on land, the white areas are covered with clouds. The FSC of more than 80% of snow area in the Arctic is >0.6; The FSC between 45–60°N is mostly concentrated at 0.1–0.5.

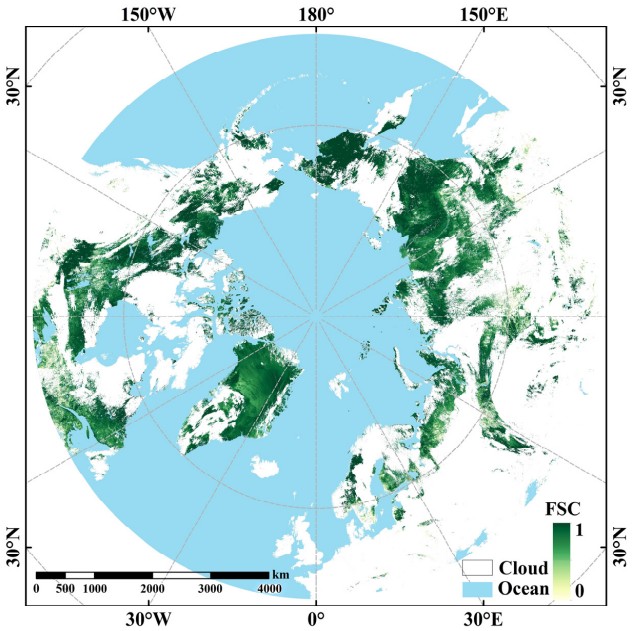

**Figure 5.** Example of FSC in Pan-Arctic (26 March 2005).

*4.2. Error Distribution of the Proportion of Snow Cover Area under the Influence of Vegetation on Different Classification*

All Landsat SR images and their corresponding MOD09GA images from 2018 to 2019 were selected, and FSC values of these two groups of images were obtained, respectively. The FSC of Landsat 8 SR was used as the TFSC to calculate the distribution of R, $R^2$, RMSE, and MAE (Figures 6 and 7). In Figure 6, the average MAE of VFSC and TFSC is calculated based on all types of underlying surfaces to illustrate the spatial distribution of MAE on different underlying surfaces.

The accuracy of comparing FSC calculated by BV-BLRM and TFSC was verified according to different land cover types. On different underlying surfaces, the average MAE, RMSE, R, and $R^2$ of the BV-BLRM model results on various underlying surfaces were 0.14, 0.17, 0.809, and 0.658. The MAE, RMSE, R, and $R^2$ of farmland were 0.07, 0.15, 0.9, and 0.81. The MAE, RMSE, R, and $R^2$ of bare land were 0.11, 0.11, 0.86, and 0.73. The forest's MAE, RMSE, R, and $R^2$ were 0.2, 0.22, 0.68, and 0.46. The shrub's MAE, RMSE, R, and $R^2$ were 0.18, 0.2, 0.7, and 0.49, respectively (Figure 7d). The degree and height of vegetation cover are highly correlated with the accuracy of FSC.

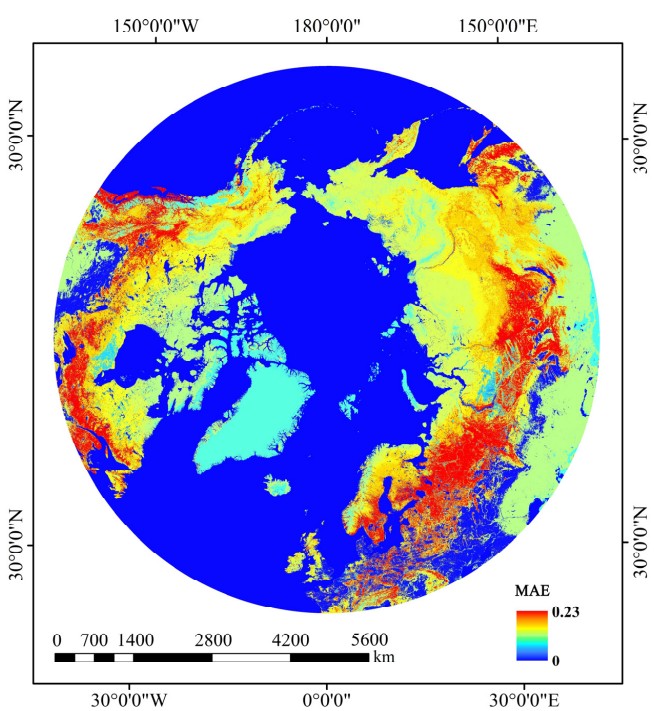

**Figure 6.** The MAE spatial distribution of FSC was calculated based on the BV-BLRM model.

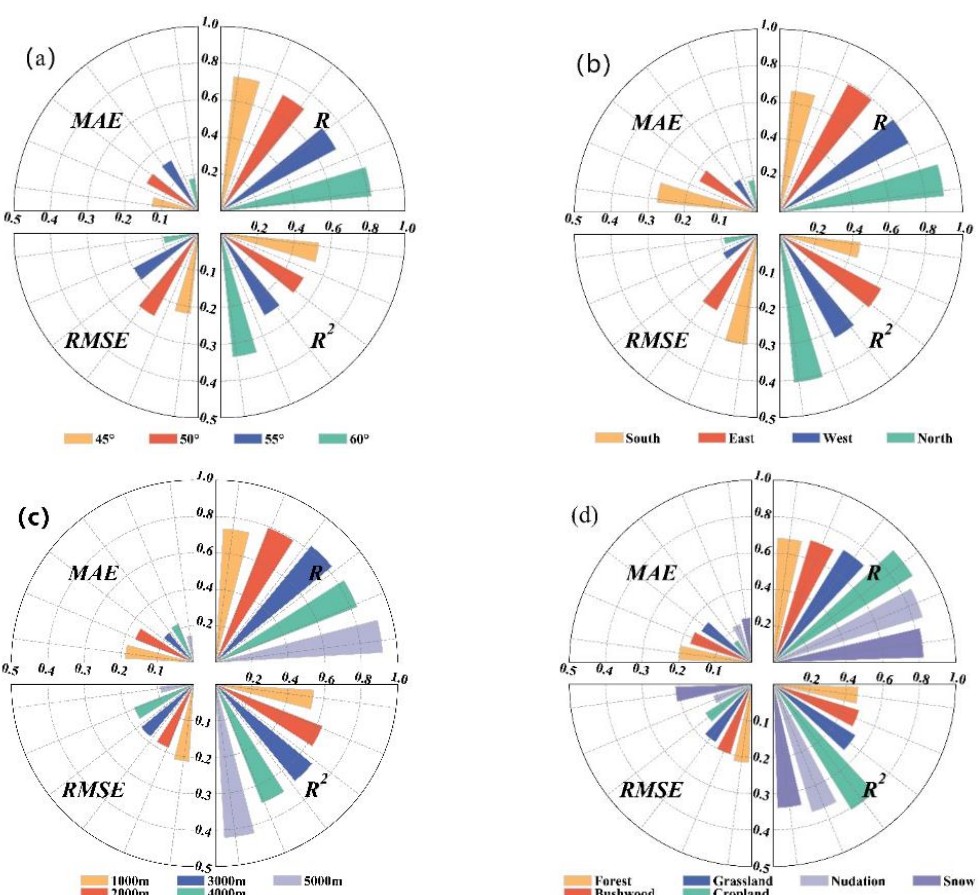

**Figure 7.** FSC (BV-BLRM) classification conditions ((**a**) latitudes by level; (**b**) different slopes; (**c**) elevations by level; (**d**) different underlay surfaces).

The vertical distribution of vegetation with altitude is shown in Figure 8; among them, the underlying surface has the broadest distribution of grassland, and the underlying surface with an altitude of 1000 m has the most abundant and relatively average types. At altitudes of 2000 m and 3000 m, the proportion of grassland and forest reaches more than 90% of the pixel number of the underlying surface of this category. In terms of the overall accuracy, the accuracy increases with the increase in DEM. The average MAE, RMSE, R, and $R^2$ of FSC calculated by BV-BLRM are 0.108, 0.159, 0.806, and 0.653, respectively. The highest accuracy is in the 5000 m area. MAE, RMSE, R, and $R^2$ are 0.07, 0.09, 0.92, and 0.84, respectively. The lowest accuracy was in the 0–1000 m region, and MAE, RMSE, R, and $R^2$ were 0.07, 0.15, 0.72, and 0.52, respectively (Figure 7c).

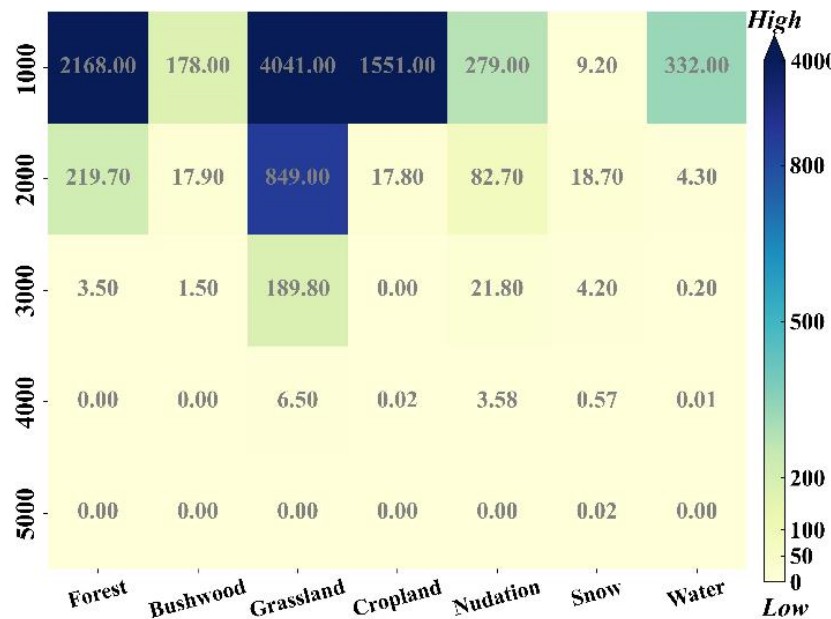

**Figure 8.** Proportion of land-cover types at different altitudes (legend represents 0 to 400‰).

The latitude is divided into 45–50°N, 50–55°N, 55–60°N, and >60°N. The accuracy is higher in the high-altitude area (>60°N). The FSC MAE, RMSE, R, and $R^2$ calculated by BV-BLRM were 0.13, 0.194, 0.759, and 0.581. The lowest accuracy was in the latitude 50–55°N region, and MAE, RMSE, R, and $R^2$ were 0.16, 0.25, 0.71, and 0.51 (Figure 7a).

Aspect is defined as the direction in which the slope normal is projected on the horizontal plane (also known as the direction of high and low). The slope aspect plays a vital role in mountain ecology—the orientation of the mountain influences sunshine duration and solar radiation intensity. For the northern hemisphere, the radiation income is highest on the southern slope, the southeastern and southwestern slopes, the eastern and western slopes, the northeastern and northwestern slopes, and least on the northern slope. According to the energy income, this study is divided into the south, east, west, and north slopes. The results show that the overall accuracy increases inversely with the energy income of each slope direction. The average MAE, RMSE, R, and $R^2$ of FSC calculated by BV-BLRM were 0.162, 0.185, 0.791, and 0.632, with the highest accuracy in the north slope, and the MAE, RMSE, R, and $R^2$ were 0.09, 0.09, 0.9 and 0.81. The south slope had the lowest accuracy, and the MAE, RMSE, R, and $R^2$ were 0.27, 0.3, 0.67, and 0.44 (Figure 7b).

### 4.3. FSC Accuracy Evaluation under Different Underlying Surface, Elevation, Aspect and Latitude

All Landsat 8 SR images in Figure 1 from 2018 to 2019 were selected to obtain the FSC and the corresponding MOD09GA images. The traditional univariate linear algorithm of MODIS was used to obtain the FSC values of the MOD09GA images. The FSC of Landsat 8 SR was taken as the TFSC, and the average R, $R^2$, RMSE, and MAE between them were calculated (Figure 9).

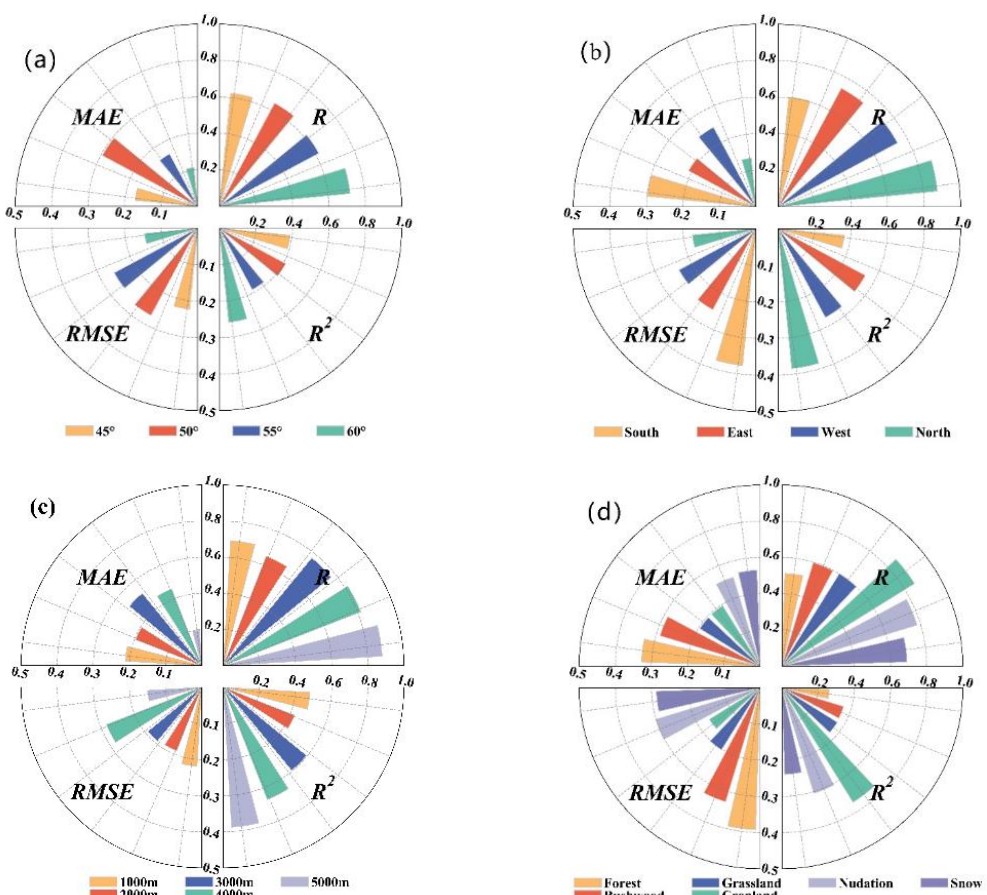

**Figure 9.** FSC (MOD-FSC) classification conditions ((**a**) latitudes by level; (**b**) different slopes; (**c**) elevations by level; (**d**) different underlay surfaces).

The results show that the highest accuracy is obtained for the north slope, DEM 5000 m, and cropland area. MAE, RMSE, R, and $R^2$ in the north slope were 0.13, 0.17, 0.87, and 0.77. The MAE, RMSE, R, and $R^2$ of DEM 5000 m were 0.01, 0.15, 0.88, and 0.77. The MAE, RMSE, R, and $R^2$ in farmland were 0.19, 0.16, 0.86, and 0.74. The lowest accuracy is 55–59° north latitude, south slope, 0–1000 m, and forest area. The MAE, RMSE, R, and $R^2$ of 55–59° north latitude were 0.16, 0.26, 0.62, and 0.38. On the southern slope, the MAE, RMSE, R, and $R^2$ were 0.29, 0.37, 0.6, and 0.36. The MAE, RMSE, R, and $R^2$ of DEM 0–1000 m were 0.17, 0.19, 0.65, and 0.42. The MAE, RMSE, R, and $R^2$ in forest area were 0.33, 0.39, 0.51, and 0.26 (Figure 9).

We calculated the R, $R^2$, RMSE, and MAE of the FSC under different classification criteria (Figure 10). The research results show that each category's accuracy has improved. The three categories with the most improvement are DEM 2000–3000 m and forest and bushwood areas, respectively, with $R^2$ increasing by 0.21 between DEM 2000 and 3000 m. The forest $R^2$ increased by 0.2. The bushwood MAE increased by 0.18. The three categories with the slightest improvement in accuracy were 60°N, DEM 5000 m, and the nudation area (Figure 10).

Combined with the DEM distribution in Figures 1 and 2 of the underlying surface distribution, compared with only a normalized index of the FSC algorithm, for the normalized difference vegetation index, this factor is introduced to the FSC accuracy mainly concentrated in the lush vegetation areas, the improvement of the accuracy and vegetation coverage are highly correlated, and the accuracy of the progress is less for year-round snow or nudation, bare ground, and other areas. Therefore, it is further proved that vegetation is the dominant factor for the FSC extraction algorithm error formation.

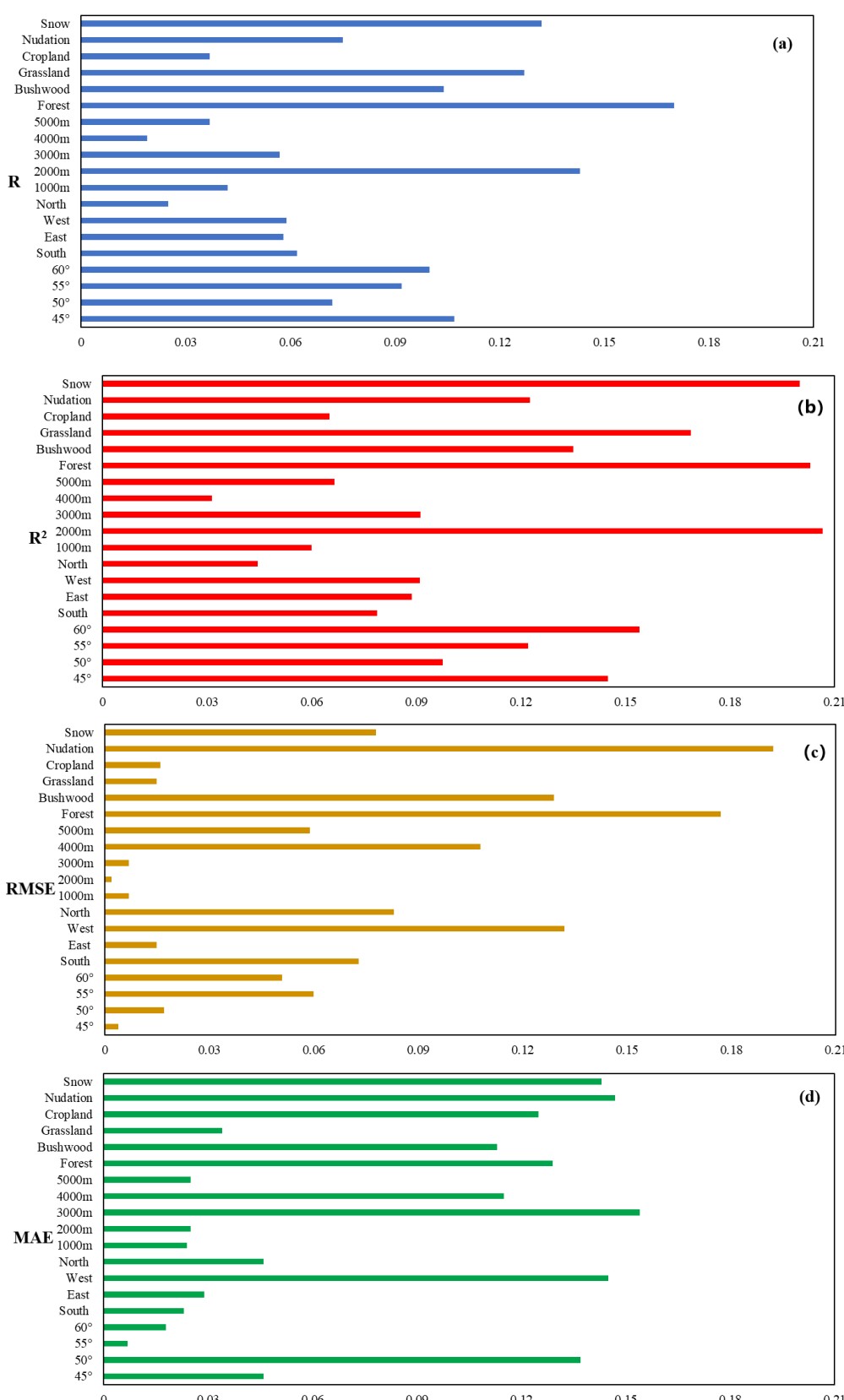

**Figure 10.** Accuracy improvement of FSC under different classification conditions ((**a**) represents the R, (**b**) represents the R$^2$, (**c**) represents the RMSE, and (**d**) represents the MAE.).

## 5. Discussion

Vegetation is an important influencing factor for almost all types of global remote sensing fractional snow cover products [36]. In this study, it was further demonstrated that among the factors, vegetation had the most significant impact on the extraction accuracy of the proportion of fractional snow cover. This conclusion is also consistent with previous studies, which showed that the decision rules and methods established by combining the normalized snow index and normalized vegetation index could improve the accuracy of snow mapping and FSC estimation in forest areas [28,32,37].

In existing studies, it is generally acknowledged that the error of the FSC in the viewable snow cover area in forest areas is large [8,25,30,38,39], which is consistent with the view of this paper. The accuracy of snow detection based on the multi-index technique is more than 90%, but it needs a lot of auxiliary data and prior knowledge. In areas with complex terrain, the accuracy of FSC based on terrain and vegetation estimation in a complex alpine forest environment can reach the highest average error of 0.0002, RMSE of 0.10, and MAE of 0.08 [20,40]. The accuracy of the FSC obtained by this method is greater than that of the BV-BLRM model. In this paper, the RMSE is 0.2 and MAE is 0.15. In the complex and heterogeneous vegetation coverage environment, Xiao et al. (2022) improved the FSC accuracy of MODIS by improving the method (using multivariate in MODIS products, 20 sub models trained by ERT method) and implementing canopy adjustment synthesis, with an RMSE of 0.124 [41]. However, the areas in the above studies are all small regional scales, and the underlying surface type is relatively singular. Moreover, the algorithm is not extended to the global scale for verification. There is a relative lack of research on global-scale FSC mapping. The machine learning algorithm is a black box process, which is incomprehensible to the physical mechanism; it requires a large number of authentic training sample data to achieve better training results [20]. Current methods mainly focus on the linear relationship between NDSI and FSC, and the overall error is large. The R, $R^2$, RMSE, and MAE are 0.62, 0.38, 0.29, and 0.21, respectively [25]. Compared with the traditional MOD-FSC, the accuracy of the BV-BLVM model is improved by 28.4% on average. In this study, the classification verification method was introduced, which classified the main underlying surface and the factors that formed the difference between the underlying surface, and the impact of vegetation on the accuracy of FSC was specifically discussed.

The snow cover observation value of high-resolution Landsat 8 is relatively more accurate than that of MODIS data [25]. In a large number of studies on FSC calculation using MODIS reflectance data, the FSC calculated from Landsat data is regarded as a "real" observation [18,23,24,42]. Therefore, in the verification, high-resolution data are used to replace the measured data. In the SNOMAP algorithm, the part with an FSC of 10–20% is not taken into account in the calculation. The algorithm can detect the snow with FSC > 0.6, and the accuracy is up to 0.98 [34]. However, the SNOMAP algorithm will have errors due to the influence of forest cover, patch snow and sensor observation conditions, resulting in the "true" FSC having errors [43–45]. In the section with results comparison, only the FSC products of MOD10A1 are compared at present. In the subsequent research, improvements can be made to the following aspects. In the FSC product, we based on MOD09GA surface reflectance data , our algorithm can't eliminate the uncertainty inherent in it, on a global scale, there are inevitably some deviations. In the model, more parameters were introduced, such as the snow particle size, snow pollutants, snow depth, mountain shadow, etc. In terms of the choice of the TFSC, uncrewed aerial vehicles (UAV) equipped with small digital cameras have many advantages. As a valuable supplement to ground and satellite observation data, UAVs have been initially applied in snow-extent estimation [27,46], which can be used to calibrate high-resolution satellite data, and the calibrated high-resolution data can be used as the TFSC. The relationship between the distribution of vegetation and altitude, latitude, aspect, and topography is relatively complex. This paper did not consider the relationship between these factors, which can be viewed in future research.

## 6. Conclusions

This paper proposes a linear regression model of the fractional snow cover according to the vegetation area and the non-vegetation area divided by the underlying surface. The influence of vegetation and altitude on the fractional snow cover extraction is quantitatively calculated. Firstly, MOD09GA data were used to extract the proportion of snow area in the study area, and Landsat 8 SR data were used to verify the model's accuracy. At the same time, it was compared with the linear model for calculating the proportion of snow area provided by the MOD10A1 snow area product. Then, the study area was verified separately by the method of classification verification, and the variation rule of the fractional snow cover in different underlying surfaces, elevation, latitude, and terrain was obtained. Get conclusions as follows:

(1) The BV-BLVM model with NDVI can better extract the proportion of FSC, and the overall accuracy is significantly improved. Compared with the traditional MODIS linear univariate algorithm, the BV-BLVM model has an average 28.4% increase.

(2) From the spatial distribution of FSC errors on different underlying surfaces, the verification results of the BV-BLRM model show that FSC errors are still relatively large when the underlying surface is covered with vegetation and are positively correlated with vegetation height and coverage. However, when the underlying surface is non-vegetation, or little vegetation cover can be regarded as bare ground, the error of FSC decreases significantly.

(3) Indirectly, altitude determines the vertical distribution of vegetation, and the slope aspect and latitude are also indirectly related to the distribution of vegetation. The verification results of the BV-BLRM model show that the FSC error of the category with a large proportion of vegetation cover and a large number of vegetation cover pixels is relatively large; otherwise, the FSC error is small.

(4) Compared with the traditional MODIS linear univariate algorithm under different classifications, the spatial accuracy of the BV-BLRM model improved the most when the underlying surface was forest, with an average of 30.5%. When the underlying surface was covered with year-round snow, the accuracy improved the least, with an average of 12.2%.

Therefore, vegetation plays an essential role in the calculation of fractional snow cover. This study is expected to improve the accuracy of global FSC estimation.

**Author Contributions:** Conceptualization, Y.M. and D.S.; Methodology, Y.M. and H.Z.; Software, H.Z. and W.J.; Formal analysis, J.W. and W.J.; Data curation, H.L.; Writing—original draft, Y.M. and D.S.; Writing—review & editing, Y.M. and D.S.; Visualization, J.W. All authors have read and agreed to the published version of the manuscript.

**Funding:** This research was funded by the Foundation from Key Laboratory of Disaster Prevention and Mitigation in Qinghai Province: QFZ-2021-G01; National Natural Science Foundation of China: 41971399; Natural Science Foundation of Qinghai Province: 2020-ZJ-731.

**Data Availability Statement:** Ma, Y., Li, H. Time series data of snow area ratio in the Arctic (2000–2019). https://doi.org/10.11888/Snow.tpdc.270330. CSTR:18406.11.Snow.tpdc.270330; Ma, Y. Global average annual snow cover proportion data (2000–2021). https://doi.org/10.11888/Cryos.tpdc.272724. CSTR: 18406.11.Cryos.tpdc.272724.

**Acknowledgments:** The authors would like to thank the editor and anonymous reviewers for their valuable comments and suggestions on this article.

**Conflicts of Interest:** The authors declare no conflict of interest.

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
