# Peer review of "Estimating Fractional Snow Cover in the Pan-Arctic Region Using Added Vegetation Extraction Algorithm"

_remotesensing, doi:10.3390/rs15030775_

Round 1

Reviewer 1 Report

Vegetation has a significant influence on the FSC estimation over forest regions. This paper developed the BV-BLRM model to calculate FSC in the pan-Arctic region by combining NDVI and NDSI. After validating against Landsat 8 data and MODIS FSC, the estimated FSC shows obvious improvements in areas with high vegetation coverage. Despite of its significance, several issues still need to be solved before a publication. The more detailed information about datasets (e.g., start and end date, data volume), the thresholds selection of NDVI and NDSI in BV-BLRM model, and the error distribution of the estimated FSC could be sufficiently explained. In addition, the English of this paper should be further refined so as to improve the overall presentation.

Other comments and suggestions:

1. Line 17, change “…developed the based normalized difference vegetation index (NDVI) Bivariate Linear Regression Model (BV-BLRM)…” to “…developed the normalized difference vegetation index (NDVI) based Bivariate Linear Regression Model (BV-BLRM)…”. What does the abbreviation BV stand for?

2. Line 18, it is not clear for your accuracy assessment strategy. What are the multiple samples and different classifications?

3. Line 39, it is suggested to provide the introduction about empirical model/semi-empirical model, mixed pixel decomposition, and machine learning models.

4. Line 105, it is not clear for the classification and data flow in Figure 1.

5. Line 113, it is suggested to directly use NDSI and NDVI in Equation 1.

6. Line156-159, the number of training datasets in the text (15 for forest area and 23 for non-forest area) is not as the same as that in Figure 2 (9). For Figure 2, change “Km” to “km”, add the unit for DEM, add the legend for red and black box.

7. Line 172, “part” means “aspect”?

8. Line 178, change “will use” to “used”.

9. Line 189, change “red light reflectance” to “reflectance of red band”.

10. Line 193, change “SNAP” to “SNOMAP”.

11. Line 196, why did you use 0.2 and 0.1 as the thresholds for NDSI and NDVI to recognize snow in the forest area?

12. Line 216-218, add the reference for Equation 12.

13. Line 221-233, which year does the land cover type data come from?

14. Line 234, the reclassification criteria for elevation in Table 1, it is suggested to mask out the regions where elevation<0 m, and to combine the regions where elevation ≥3000 m into one category in your study.

15. Line 247, why did you use 0.2 as the NDVI threshold to define vegetation cover?

16. Line 248, change “bare ground” to “non-vegetation”.

17. Line 258, it is suggested to move the Table 2 to section 3.2.

Author Response

We greatly appreciate your careful revision of our manuscript.

For these questions, we will answer them in detail later. We have examined the syntax and structure of the paper in detail. In addition, the English grammar editing has been completed. Details are in the appendix (Response to Anonymous Review #1).

Reviewer 2 Report

This manuscript deals with the estimate of fractional snow cover in the pan-Arctic region using BV-BLRM model. The authors developed the model to estimate FSC using many remote sensing data and tried to show its better performance in the manuscript. However, I think that the manuscript would be accepted after major revision.

Comments

1. Line 90: Calculate NDVI --> Calculation of NDVI

2. Line 144: Authors had better describe the definition of invalid values and how to remove them for better understanding.

3. Line 152: land area north --> land area above north

4. Figure 2: The authors described forest and non-forest in Lines 152 to 153. But, I could not find the difference in Figure 2.

5. Lines 169-171: I could not understand this paragraph. Thereofore, the authors need to repharase this.

6. Line 165: The authors had better describe the method how to remove the effect of the atmosphere.

7. Line 177: azimuth Angle --> azimuth angle

8. Lines 241-242: This sentence is nor clear to me. The authors need to rephrase it.

9. Line 246: The authors need to describe the method how to remove the cloud pixels.

10. Lines 266-267: Confirm the font style.

11. In Table 2: validates --> Validation of

12. Line 287: Figure 6 below --> Figure 6

13. Section 4.3: I expect that the authors would show the results of BV-BLRM for different vagetation. However, I think that the authors showed the result from MODIS in this Section. The authors need to clarify this.

Author Response

We greatly appreciate your careful revision of our manuscript.

Details are in the appendix (Response to Anonymous Review #2).

Reviewer 3 Report

Major comments:

1.     What makes your algorithm perform better than MODIS standard method (Eq. 12) with respect to the linear-regression type methods?

2.     As far as I know, there are a lot of experiential regression method. What’s your mainly contribution to the fractional snow cover estimation researches? Through your publication, I didn’t find it, please show me.

3.     If you think that considering the NDVI is your major innovation, I think this study would be not good enough to publish in here. I don’t know, please give more modification to your “excellent work”.

4.     As said in “This study is helpful to improve the FSC accuracy in areas with high vegetation coverage”, please conduct some additional experiments to prove that your model has excellent capability to retrieve fractional snow cover in high vegetation cover regions.

5.     When I read your Abstract and the Conclusion, these statements are so comment and boring. I think you should give deeply revision in these two parts.

6.     How do you consider the vegetation effects in fractional snow cover estimation (Xiao et al., 2022 JAG)? Vegetation effects is so complicate in snow cover area related study, but I did see similar statements in your paper. I suggest that you must add these comments to the discussion section.

7.     You have so many training samples ("3.4 million effective snow pixels"). I have a question, why don’t you use the machine learning method? And there are a lot of methods in estimating fractional snow cover, linear empirical regression, linear spectral unmixing method and, the machine learning method (Xiao et al., 2022), in the comparison section you only compared with the MODIS standard method, why not with other method? Are there other studies that demonstrate that the Bivariate Linear Regression Models are better than the spectral unmixing methods or machine learning method?

Minor comments:

1)     Line 115: "m is the NDVI value that distinguishes forest area and non-forest area". I cannot understand it. Typically, we used land cover data to classify forest or non-forest type. Please provide some evidence that can accurately distinguish forest and grass lands.

2)     Line 126: “…the quality of sample data should be higher” How to understand it? Please rephrase this sentence.

3)     Line 127 “Traditional verification methods need to select images for verification manually, and the number of samples available for verification is limited.” It is not the TRUE!!!

4)     Line 129 please check the English language grammar. It is hard to follow!

5)     Why do you use both metrics correlation coefficient (R) and coefficient of determination (R2)? As far as I know, they only use one of them in most publications. Do you know the difference of R and R2? It is meaningless if you used both of them.

6)     Please switch the order of Section 3 and Section2.

7)     Line 193: Change “SNAP” to “SNOMAP”

8)     Line 221-223. Please rephrase it.

9)     Line 247: Please give the evidence of selecting “0.2”.

10)  Your validation only focused on several Landsat scenes. How do you calculate MAE in Arctic region (Fig. 4)?

11)  Fig. 6: The accuracy metrics is correlated with the number of samples. Please add the sample size for different altitude and land cover types.

Author Response

We greatly appreciate your careful revision of our manuscript.

Details are in the appendix(Response to Anonymous Review #3).

Round 2

Reviewer 1 Report

General comments:

More detailed information about datasets, the thresholds selection of NDVI and NDSI in BV-BLRM model, and the error distribution of the estimated FSC are provided in this revised version. Just a few comments:

Comment 1:

L105, change “samples” to “sample areas”.

Comment 2:

L109-110 and L111-112 are repeated.

Comment 3:

L424, change “Xiao” to “Xiao et al. (2022)”.

Author Response

We greatly appreciate your valuable suggestions and comments. Details of the modifications are in the attachment (Response to Review 1).

Reviewer 2 Report

The authors modified the manuscript according to the comments. Therefore, I would like to recommend this article be accepted in its present form.

Author Response

We greatly appreciate your valuable opinions and comments, which have greatly improved the quality of the article.

Reviewer 3 Report

All questions have been addressed

Author Response

(The authors gave the same response as above.)
